# An Overview of Diet and Physical Activity for Healthy Weight in Adolescents and Young Adults with Type 1 Diabetes: Lessons Learned from the ACT1ON Consortium

**DOI:** 10.3390/nu15112500

**Published:** 2023-05-27

**Authors:** Franziska K. Bishop, Ananta Addala, Karen D. Corbin, Franklin R. Muntis, Richard E. Pratley, Michael C. Riddell, Elizabeth J. Mayer-Davis, David M. Maahs, Dessi P. Zaharieva

**Affiliations:** 1Division of Endocrinology, Department of Pediatrics, School of Medicine, Stanford University, Stanford, CA 94304, USA; 2AdventHealth, Translational Research Institute, Orlando, FL 32804, USA; 3Department of Nutrition, University of North Carolina at Chapel Hill, Chapel Hill, NC 27599, USA; 4School of Kinesiology and Health Science, Muscle Health Research Centre, York University, Toronto, ON M3J 1P3, Canada; 5Stanford Diabetes Research Center, Stanford, CA 94305, USA

**Keywords:** type 1 diabetes, exercise, physical activity, continuous glucose monitoring, overweight, weight maintenance, glycemic control, obesity

## Abstract

The prevalence of overweight and obesity in young people with type 1 diabetes (T1D) now parallels that of the general population. Excess adiposity increases the risk of cardiovascular disease, which is already elevated up to 10-fold in T1D, underscoring a compelling need to address weight management as part of routine T1D care. Sustainable weight management requires both diet and physical activity (PA). Diet and PA approaches must be optimized towards the underlying metabolic and behavioral challenges unique to T1D to support glycemic control throughout the day. Diet strategies for people with T1D need to take into consideration glycemic management, metabolic status, clinical goals, personal preferences, and sociocultural considerations. A major barrier to weight management in this high-risk population is the challenge of integrating regular PA with day-to-day management of T1D. Specifically, exercise poses a substantial challenge due to the increased risk of hypoglycemia and/or hyperglycemia. Indeed, about two-thirds of individuals with T1D do not engage in the recommended amount of PA. Hypoglycemia presents a serious health risk, yet prevention and treatment often necessitates the consumption of additional calories, which may prohibit weight loss over time. Exercising safely is a concern and challenge with weight management and maintaining cardiometabolic health for individuals living with T1D and many healthcare professionals. Thus, a tremendous opportunity exists to improve exercise participation and cardiometabolic outcomes in this population. This article will review dietary strategies, the role of combined PA and diet for weight management, current resources for PA and glucose management, barriers to PA adherence in adults with T1D, as well as findings and lessons learned from the Advancing Care for Type 1 Diabetes and Obesity Network (ACT1ON).

## 1. Introduction

Overweight and obesity among people with type 1 diabetes (T1D) parallel the increasing trends in the U.S. population [1,2]. Among young adults (18–25 years) in the T1D Exchange Registry, 31% were overweight and 15% were obese [3]. Young adults in general, particularly those that are already overweight, also experience the highest risk of weight gain relative to other age groups [4]. Longitudinal studies show that body adiposity increases in people with T1D as they age [5,6]. Obesity is now accepted as an important contributor to long-term cardiovascular disease (CVD) risk in people with T1D [7,8,9]. This was clearly demonstrated by Purnell and colleagues [10] whereby the participants in the intensive insulin therapy intervention group from the Diabetes Control and Complications Trial (DCCT) gained the most weight and ultimately experienced the same CVD risk as the control group but also had less weight gain. While there is no single diet or specific eating pattern for people with T1D, metabolic status, clinical goals, and personal preferences, and sociocultural factors should always be considered [11]. In addition, existing evidence shows no ideal macronutrient distribution for weight loss [12] and diet plans should be individualized for all youth with diabetes. Carbohydrate/energy intake recommendations in older, overweight, or obese adolescents may be lower (~40% energy) with higher protein intake (~25% energy) according to clinical consensus [12]. There is longstanding observational evidence supporting a range of health benefits from regular physical activity (PA) and exercise among individuals with T1D [13,14,15], including increased life expectancy [16,17]. In addition to reducing CVD risk, regular PA and exercise are also critical to maintaining overall health and fitness [15,18]. Increased levels of PA have also been associated with reductions in diabetes-related complications, such as neuropathy, nephropathy, and retinopathy [19]; improved psychosocial well-being [20]; and overall quality of life [21]. The importance of PA in the treatment and prevention of obesity is well documented in the general population [22,23], yet the role of PA in weight loss and weight management for T1D has not yet been explored. The role of combined PA and diet for weight management in T1D is also underexplored. Barriers to PA and diet in those with TID include less metabolic flexibility during exercise [24,25,26,27,28], fear of exercise-associated hypoglycemia [29], and lack of specific recommendations for weight management for people with T1D [30]. However, with a lack of evidence and research on obesity treatment in people with T1D, there is currently limited evidence to base recommendations on [30].

### 1.1. ACT1ON Overview

In June 2015, ACT1ON, Advancing Care for Type 1 Diabetes and Obesity Network [31,32], was established, a consortium comprising of transdisciplinary scientists at three leading institutions, University of North Carolina at Chapel Hill (UNC), AdventHealth Translational Research Institute (TRI), and Stanford University. ACT1ON completed a pilot trial of dietary strategies to facilitate weight loss in young adults with T1D (DP3DK113358; NCT03651622). The trial used a fully remote protocol to co-optimize glycemic management and weight status. ACT1ON has conducted studies related to T1D and PA, safe exercise among individuals with T1D, and execution of diabetes self-management, lifestyle, and weight management in T1D [30,33,34]. While ACT1ON has primarily focused on dietary strategies in T1D, sustainable weight management generally requires the combination of diet and regular PA. These approaches must be optimized toward the underlying metabolic and behavioral challenges unique to T1D to support glycemic management throughout the day. The next steps and future work with the ACT1ON consortium include building upon our dietary modification clinical trial for weight management in T1D [30,33] and a pilot and feasibility trial of a behavioral intervention to safely increase PA levels among overweight and obese young adults with T1D, combined with a tailored diet to promote weight loss. Future research focusing on the efficacy, acceptability, feasibility, and implementation of tailored weight management interventions in youth and adults with T1D is also warranted.

### 1.2. Role of Combined Diet and PA for Weight Management in T1D

Weight management in T1D is a complex topic, and there continue to be substantial gaps in knowledge in this population. Individuals with T1D can have alterations in energy metabolism during rest and exercise because of exogenous insulin therapy. In general, less metabolic flexibility is observed during exercise in those with T1D, with a blunted capacity to oxidize exogenous carbohydrates and endogenous lipids when blood glucose levels are elevated [24,25,26,27,28]. These impairments in metabolic flexibility may, in turn, impact the capacity to lose weight with regular PA [35]. However, weight loss can be achieved with regular training, particularly if small reductions in total daily insulin dosages occur [36] and excessive energy intake to prevent and/or treat hypoglycemia is generally avoided [37]. Unfortunately, unhealthy and potentially dangerous weight loss practices, such as skipping insulin doses, excessive fasting, vomiting, and the use of laxatives in T1D are common strategies if weight loss is desired [38].

There is no single diet or specific eating pattern that all individuals with T1D should follow; however, based on consensus guidelines, it is recommended to emphasize consuming non-starchy vegetables, minimal sugar, and refined grains, and choose whole foods over highly processed ones [11]. According to the American Diabetes Association (ADA), individualization of dietary recommendations is endorsed to account for metabolic status, clinical goals, personal preferences, and sociocultural considerations [11]. Additionally, it is well recognized that in clinical practice, when a given intervention does not produce the desired outcome in a reasonable amount of time, a different approach can be taken. Traditionally, in randomized controlled trials (RCTs), regardless of whether an individual is responding to the outcome in the desired manner, individuals must remain in the condition to which they were randomized throughout the duration of the trial. In recent years, the importance of tailoring interventions within the context of rigorously controlled clinical trials has been recognized, and a variety of designs have emerged to this end [39]. The ACT1ON consortium is the first to simultaneously address glycemic management and weight loss (DP3DK113358) using a Sequential Multiple Assignment Randomized Trial (SMART). This trial reported significant weight loss among both a low-fat and low-carbohydrate approach, with no detrimental effects on glycemic management among young adults with T1D and overweight or obesity [33]. Although it is apparent that PA is an essential component of weight maintenance following weight loss [40]; studies investigating the combined effects of PA and diet in T1D are still limited [34]. As such, research studies aimed at assessing the effects of diet alone, PA alone, and the combination of diet plus PA on weight management in T1D are warranted.

### 1.3. Current Resources for PA and Glucose Management in T1D

Several factors may influence the trend and magnitude of glucose changes during and following PA in adults and youth with T1D (Figure 1). In general, the key contributors to glycemic responses to PA appear to include the type of PA (aerobic vs. resistance or mixed activity), prandial status (fed vs. fasted activity), circulating insulin level, blood glucose level at the start of the activity, glucose trends preceding the activity, composition of the most recent meal or snack, as well as the intensity and duration of the activity [41,42]. Recent evidence suggests that the time of day at which exercise is performed and prandial status are important determinants for the glucose response to PA [43,44,45,46,47,48,49]. Specifically among those with T1D, exercise performed later in the day has been shown to promote a greater drop in glucose levels compared to morning exercise, with afternoon and late-day exercise also heightening the risk of nocturnal hypoglycemia [46,50]. Alternatively, performing moderate intensity aerobic exercise in the morning and in a fasted state appears to have a lower risk of exercise-associated hypoglycemia and improved glucose time in range (TIR) during the 36 h following activity compared to the same exercise performed in the afternoon [45]. However, performing higher bursts of activities, such as explosive resistance exercise or high-intensity interval training (HIIT), specifically in a fasted state in the morning, may promote more less of a drop in glycemia and in some cases, even lead to hyperglycemia [43,44,51]. This may be in part due to differences in substrate oxidation during exercise performed in a fasted versus postprandial state, or may be due to higher levels of circulating cortisol and growth hormone in the morning [46,52].

While the time of day at which exercise is performed can be largely driven by an individual’s schedule, personal preferences, or chronotype, understanding the role of the time of day on the glycemic response to exercise may benefit individuals with T1D by guiding exercise selection. In addition, the type or intensity of exercise performed may also lead to variable glycemic responses for individuals with T1D [41]. Specifically, moderate intensity aerobic exercise tends to promote a decrease in blood glucose levels and anaerobic exercise, such as HIIT or resistance exercise (particularly in the fasted state), tends to cause a rise in glycemia during the activity, or promote a moderating effect on glycemia [18,48]. Therefore, individuals with T1D that have concerns about hypoglycemia around exercise may consider choosing more anaerobic forms of exercise in the fasted state to potentially reduce the risk of hypoglycemia during and following activity.

Continuous Glucose Monitoring (CGM) systems are also important to discuss as widely-used and powerful tools in diabetes management and based on A-level evidence in ADA practice guidelines [53] and CGM position statements [54]. By providing continuous glucose tracings and trends around exercise, CGM systems can be used as a tool to assist with glucose management, inform exercise strategies, and improve patient safety and confidence around exercise [55]. Moser et al. [54] describe safe glycemic ranges for exercise and how sensor glucose readings in addition to directional trend arrows can be used to better inform individuals with T1D on managing glucose levels before, during, and after exercise. It is also important to take into consideration the potential increase in lag time (i.e., delay) with CGM interstitial glucose readings commonly seen during rapid changes in glucose during or post-exercise [56,57,58,59]. To account for some of this discrepancy, more frequent fingerstick blood glucose monitoring is recommended around exercise for all individuals with diabetes [56,57]. The use of CGM devices can also be a tool to help inform when fast-acting carbohydrates should be consumed to prevent or treat hypoglycemia. However, with the potential increase in CGM lag-time around activity, setting higher alerts around hypoglycemia or proactively treating hypoglycemia may be strategies implemented during PA for individuals with T1D [54].

These and other individual considerations are critical to consider for safe engagement in PA [41,60]. Strategies for PA management need to be tailored to each individual because, despite making exercise-specific adjustments for care, many individuals with T1D still report significant difficulties with glycemic management as it relates to PA and planned exercise [61]. It is imperative to systematically study behavioral strategies to promote participation in PA among individuals with T1D that will safely support appropriate glycemic excursions and address barriers to regular adherence to PA guidelines, such as those described above.

### 1.4. Integrating Motivational Interviewing and Problem-Solving Skills Training to Address Barriers to PA Adherence

To provide much-needed support to increase overall PA levels and adherence to PA among individuals with T1D, ACT1ON draws upon the conceptual framework developed by our team to address diabetes self-management [62]. The Health Belief Model, Transtheoretical Model, Theory of Reasoned Action [63,64], and analysis and integration of theory in social and health psychology [65,66] are all established theories of health behavior that provide a conceptual framework positing information, motivation, and skills as necessary for behavior change. Our application of this theory [62] (Figure 2) integrates motivational interviewing (MI) and problem-solving skills training (PSST) to teach practical problem-solving tailored to the patient by way of a multi-faceted intervention, supplemented by a flexible array of tools (e.g., optional participant-defined cell phone reminders, educational materials) designed to target pragmatic barriers to adherence (e.g., fear of hypoglycemia). Once motivation for change is achieved, PSST, a systematic approach to problem-solving, is taught to participants to help them make the behavioral changes they desire. This behavioral approach has been shown to improve Diabetes Self-Management Profile—Self Report (DSMP-SR) and psychosocial outcomes in T1D, including motivation and problem solving [67]. ACT1ON participants met with Registered Dietitians (RD) at the intervention introduction in-person or on Zoom monthly, as well as phone check-ins utilizing the above-described theories and approaches. Additional sessions with the RD also occurred during diet re-randomization at 3 and 6 months [33] (Figure 3).

### 1.5. What Is Needed—Gaps in the Literature

Overall, there is a pressing need to generate a scientific evidence base to develop weight management guidelines specific to individuals living with T1D [31,68]. Despite known health benefits, the majority of individuals with T1D do not engage in the recommended amount of at least 150 min of moderate-to-vigorous physical activity (MVPA) per week, with no more than two consecutive days without PA [69,70,71,72]. Thus, a tremendous opportunity exists to improve PA participation and adherence to regular PA and exercise in this population. Individuals with T1D and healthcare professionals have identified numerous challenging aspects of diabetes management, including having limited knowledge around exercise effects on glycemia [73,74]. In addition, it is estimated that ~50–60% of people with T1D choose not to participate in regular PA because of the associated risks of dysglycemia, including fear of hypoglycemia, the loss of diabetes control, busy work schedules, and low fitness levels [75]. A main barrier to weight management in this high-risk population is the challenge of integrating regular PA with the day-to-day management of T1D [76]. Despite relatively low participation in PA, it is recommended that all individuals with T1D engage in regular PA and reduce sedentary behavior [77,78]. Combined PA and diet interventions to simultaneously address glycemic and weight management are critically needed to help ensure the long-term health of individuals with T1D.

## 2. Published ACT1ON Study Results

The ACT1ON SMART pilot analyses were conducted on data collected up to the temporary close of data collection due to COVID-19, termed “pre-COVID-19” data [33]. In summary, young adults aged 19–30 years with T1D ≥ 1 year with a body mass index of 27–39.9 kg/m^2^ participated in a 9-month SMART pilot study [33]. Re-randomization occurred at 3 and 6 months if the assigned diet was not deemed acceptable by study participants or effective (<2% weight reduction, HbA1c increase >0.5%, and self-reported increased or problematic hypoglycemia) [33]. From these pre-COVID-19 data, focusing on the three defined hypocaloric experimental diets studied, 38 young adults with T1D completed the first of three 3-month-long experimental diet periods, hypocaloric low-fat (30% kcal) diet based on the Look AHEAD Study [79]; hypocaloric low-carbohydrate (20–25% kcal); and the Mediterranean diet based on the PREDIMED trial [80]. Complete ACT1ON study design and methodology have been previously published [30]. Significant weight loss was observed with no difference between dietary strategies and no worsening of glycemic management. Those assigned to the Mediterranean diet (that did not require calorie restriction) had the same weight loss as the hypocaloric low-fat and hypocaloric low-carbohydrate diets. Additionally, only the Mediterranean diet resulted in a statistically significant reduction in HbA1c levels. Significant weight loss with all three diets might suggest that carbohydrate restriction may not be required for effective weight and glycemic management [33]. At the end of the 3-month diet period, a priori decision rules as part of our SMART design were applied to determine whether each participant would be re-randomized. The decision rules included specific cut-points related to weight loss, hypoglycemia, HbA1c, and diet acceptability. Of note, despite the overall success in weight loss, many participants (58%) met criteria for re-randomization to a different diet. Of those, 77% failed to meet the weight loss criteria (loss of ≥2% of body weight) and 59% found their diet to be unacceptable [33]. These findings confirm what is well-recognized clinically, which is that “one size does not fit all” and the heterogeneity of weight loss responses to diet for people with T1D are complex and likely influenced by genetic, metabolic, behavioral, and environmental factors [33]. However, this trial did demonstrate that short-term weight loss is achievable for those with T1D while sustaining or improving HbA1c without increasing hypoglycemia [33]. In terms of behavioral intervention fidelity, we have previously demonstrated high and consistent adherence to intervention content and behavioral counseling strategies [67].

### Existing PA Data in T1D

Riddell and colleagues [41] published an evidence-based consensus statement for PA in adults with T1D, including detailed recommendations for suggested pre-exercise blood glucose levels, as well as insulin dosing strategies to support safe participation in planned exercise which have been summarized here as a table (Table 1).

Zaharieva et al. [81] found that among open loop pump users, a 50–80% basal insulin reduction set 90 min in advance of aerobic activity can help reduce the likelihood of hypoglycemia during exercise. To address the phenomenon of post-exercise hyperglycemia in the face of fasted, HIIT in the morning, Aronson et al. [82] found that a 50–150% insulin dose correction after exercise, based on the glucose level measured post-exercise and the participant’s usual insulin correction factor is safe and effective in adults with T1D [83]. Another recent study found increased glucose TIR (70–180 mg/dL/3.9–10.0 mmol/L) and time below range (TBR < 70 mg/dL/3.9 mmol/L) in adults with T1D in the 24 h after exercise days (instructional study videos were used), as compared to sedentary days [84]. However, in an ancillary pilot study from the parent ACT1ON study, secondary analyses looking at the relationship between MVPA and glycemia in adults showed worsened glycemia (increased time above range [TAR > 180 mg/dL/10.0 mmol/L] and decreased TIR [70–180 mg/dL/3.9–10.0 mmol/L]) on the day following reported PA with increased MVPA [34]. This variability further exemplifies the need for more research in PA interventions to simultaneously address glycemic and weight management in T1D and a need for PA education for this unique population.

## 3. Future Directions and Future Steps Ongoing Challenges

Rates of obesity and associated comorbidities in individuals with T1D are rising. Unique barriers for participation in regular PA exist for individuals with T1D, including fear of hypoglycemia as a leading barrier [75]. Further research is needed to address behavior change and barriers for individuals with T1D to participate in regular PA, while also targeting weight loss with evidence-based dietary strategies. There are no studies, to our knowledge, that have addressed strategies to facilitate participation in PA for individuals with T1D in the context of a hypocaloric diet. This is important because both weight management and PA are known to be important factors in CVD risk mitigation, and individuals with T1D are at higher risk for CVD than the general population [10]. In addition, PA is critical for long-term maintenance of weight loss. Several topics remain to be explored for the purpose of supporting increased PA levels in individuals with T1D. First, new educational platforms using patient-led peer support to facilitate PA for weight loss or weight maintenance, in collaboration with healthcare professionals, may offer more effective engagement and adherence to PA than current practices, but the evidence for this approach is lacking [85,86,87]. Second, the use of technologies to facilitate improved glucose management during and after exercise, such as the use of Automated Insulin Delivery (AID) systems [88], glucagon therapy for exercise [89] and PA advisors [90] may help to reduce barriers to PA in individuals with T1D. Third, the interactions among dietary options and exercise training for weight loss and exercise performance in T1D also need to be explored. Lastly, increasing PA can lead to increased hypoglycemia risk in individuals with T1D; therefore, glycemic management and addressing challenges around exercise should be a key part of any intervention for people living with T1D.

To address some of these gaps, RCTs and additional clinical studies are needed to test the effectiveness of increasing PA with a tailored hypocaloric diet and glycemic management in people with T1D. This work will be critical to advance work to improve how PA and diet are implemented into clinical practice to support weight loss, glycemic management, CVD risk mitigation, and quality of life for individuals with T1D and overweight or obesity.

## 4. Conclusions

It is critical to develop scientifically proven strategies and guidelines that simultaneously address glycemic maintenance, weight maintenance/loss, and safe PA to prevent both short- and long-term complications and increase well-being for individuals with T1D. With the prevalence of obesity and overweight in adults with T1D that parallels that of the general population [1,2], there is a pressing need to generate a scientific evidence base to develop weight management guidelines specific to T1D. There is no “one size fits all diet” or specific eating pattern for people with T1D and diet plans should be individualized to the person with TID [11,12]. ACTION did demonstrate that short-term weight loss is achievable for those with T1D while sustaining or improving HbA1c without increasing hypoglycemia [33]. Understanding the combined effect of T1D and obesity is a critical point for understanding which components are most effective to further support implementation and weight loss among individuals with T1D. In summary, adherence and engagement in regular PA and glycemic management around PA continue to be a challenge for many individuals with T1D and strategies need to be further developed and tailored to address weight management in this high-risk population.

## Figures and Tables

**Figure 1 nutrients-15-02500-f001:**
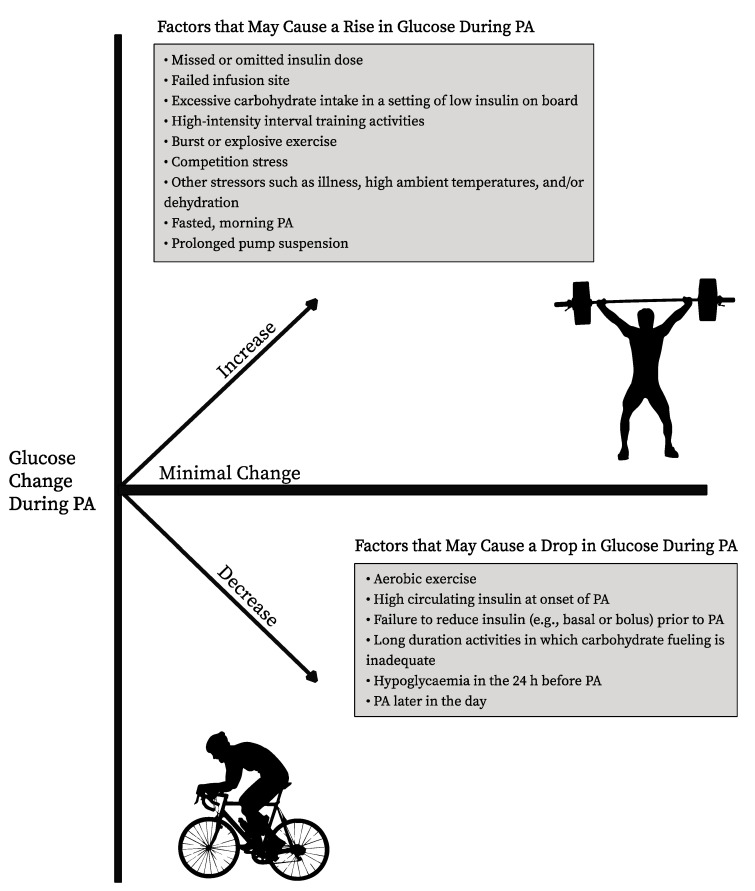
Factors likely to influence the trend and magnitude of glucose change during exercise in adults and youth with type 1 diabetes (T1D).

**Figure 2 nutrients-15-02500-f002:**
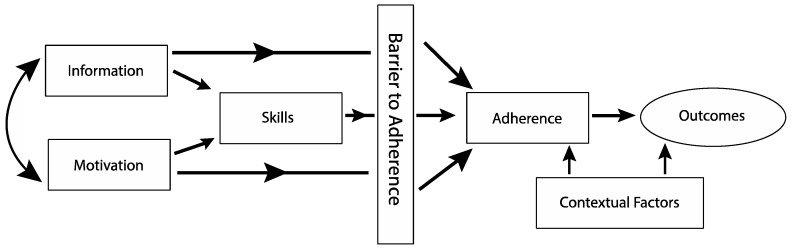
Conceptual Framework integrates motivational interviewing (MI) and problem-solving skills training (PSST) to teach practical problem-solving tailored to the patient by way of a multi-faceted intervention, supplemented by a flexible array of tools (e.g., optional participant-defined cell phone reminders, educational materials) designed to target pragmatic barriers to adherence (e.g., fear of hypoglycemia). Adapted from Kichler et al. [62].

**Figure 3 nutrients-15-02500-f003:**
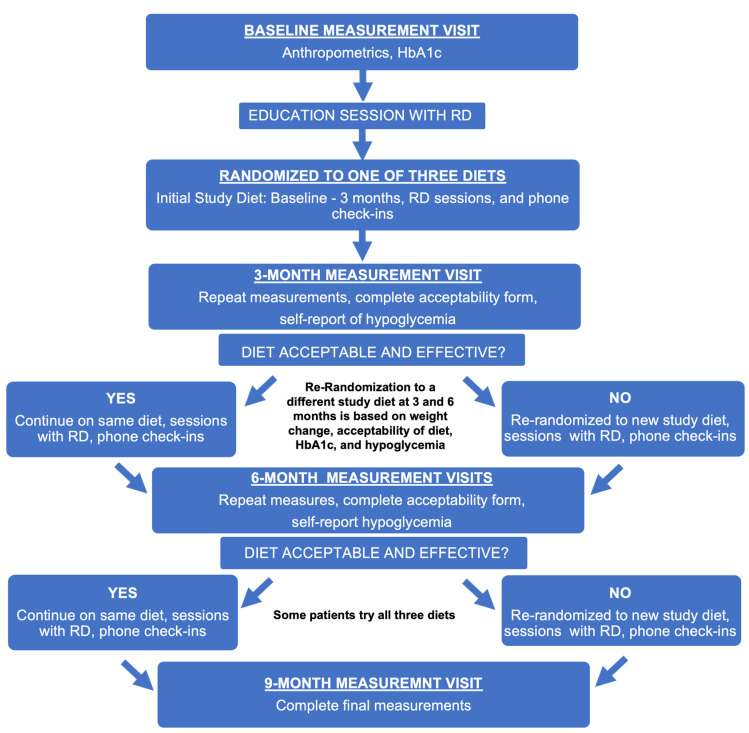
Flowchart of ACT1ON study activities, including randomization scheme, measurement visits, and education sessions with a Registered Dietitian (RD). Adapted from Corbin et al. [30].

**Table 1 nutrients-15-02500-t001:** Suggested starting blood glucose concentrations for exercise. Table modified from the exercise consensus guidelines in adults with type 1 diabetes (T1D) by Riddell et al., Lancet Diabetes Endocrinol, 2017. ↑ indicates a potential rise in blood glucose and ↓ indicates a potential drop in blood glucose.

Blood Glucose Level	Aerobic/Low Intensity	Anaerobic/High intensity
<90 mg/dL (<5.0 mmol/L)	Major hypoglycemia risk ~10–20 g carbohydrates and re-check before starting Consider insulin adjustments	May be OK to start if predictable rise seen before activity ~10–15 g carbohydrates
90–124 mg/dL (5.0–6.9 mmol/L)	~10 g carbohydrates, then start Consider insulin adjustments	OK to start
126–180 mg/dL (7.0–10.0 mmol/L)	OK to start	OK to start, but blood glucose may ↑
180–270 mg/dL (10.0–15.0 mmol/L)	OK to start, but performance may ↓	OK to start, but performance may ↓ Blood glucose may ↑ further
>270 mg/dL (>15.0 mmol/L)	If unexplained high, check ketones If small-to-moderate levels, then light intensity OK Consider 50% correction bolus	Avoid exercise

## Data Availability

The data are not publicly available in accordance with the consent provided by participants on the use of confidential data.

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
