# Peer review of "An Overview of Diet and Physical Activity for Healthy Weight in Adolescents and Young Adults with Type 1 Diabetes: Lessons Learned from the ACT1ON Consortium"

_nutrients, 2023, doi:10.3390/nu15112500_

Round 1

Reviewer 1 Report

Thank you for this excellent publication highlighting the importance of physical activity in individuals with Type 1 Diabetes and the difficulties in adherence.

A figure that summarises the results that show the numbers of participants within each arm perhaps together with a flow chat for after intervention would be beneficial.

Although the authors have highlighted relevant points in particular the difficulty in behaviour change, this has not really been discussed in detail here with information from ACT1ON as the lessons learned from the study. I think this is quite interesting and would help highlight the framework shown in Figure 2 by using information gained from ACT1ON.  For example, was motivation change achieved in all individuals? I assume that even if motivation changes the fear of hypoglycaemia is still very present (much like trauma) or is able to creep back reducing compliance to PA (as suggested by the increase in obesity in individuals with T1D). Is there continued face to face/telehealth counselling beyond self reading educational materials that participants used in ACT1ON? Or is this what is needed.

Thank you for this enjoyable read.

Author Response

May 17, 2023

Manuscript ID: nutrients-2382218
Type of manuscript: Article
Title: An overview of diet and physical activity for healthy weight in 
adolescents and young adults with type 1 diabetes: Lessons learned from the 
ACTION consortium
Authors: Franziska K Bishop *, Ananta Addala, Karen D. Corbin, Franklin R. 
Muntis, Richard E. Pratley, Michael C. Riddell, Elizabeth J Mayer-Davis, 
David M Maahs, Dessi P. Zaharieva
Received: 19 April 2023
E-mails: fbishop@stanford.edu, aaddala@stanford.edu, 
karen.corbin@adventhealth.com, frmuntis@email.unc.edu, 
richard.pratley.md@adventhealth.com, mriddell@yorku.ca, 
mayerdav@email.unc.edu, dmaahs@stanford.edu, dessi@stanford.edu
Submitted to section: Pediatric Nutrition,
https://www.mdpi.com/journal/nutrients/sections/Pediatric_Nutrition
Nutrition Managing in Pediatric Diabetes: Aspects and Challenges
https://www.mdpi.com/journal/nutrients/special_issues/pediatricdiabetes_nutrition

Responses to Reviewers

Reviewer 1:

Thank you for this excellent publication highlighting the importance of physical activity in individuals with Type 1 Diabetes and the difficulties in adherence.

A figure that summarises the results that show the numbers of participants within each arm perhaps together with a flow chat for after intervention would be beneficial.

Although the authors have highlighted relevant points in particular the difficulty in behaviour change, this has not really been discussed in detail here with information from ACT1ON as the lessons learned from the study. I think this is quite interesting and would help highlight the framework shown in Figure 2 by using information gained from ACT1ON.  For example, was motivation change achieved in all individuals? I assume that even if motivation changes the fear of hypoglycaemia is still very present (much like trauma) or is able to creep back reducing compliance to PA (as suggested by the increase in obesity in individuals with T1D). Is there continued face to face/telehealth counselling beyond self reading educational materials that participants used in ACT1ON? Or is this what is needed.

Thank you for this enjoyable read.

Response:

Thank you for reviewing this manuscript and for the thoughtful and helpful comments. We have added content to the revised manuscript to summarize some of the key ACT1ON pilot findings in Part 2, page 7 as well as a new Figure (added text and Figure below). We would also like to refer all readers to the already published pilot results manuscript in Diabetes Obesity and Metabolism (PMID: 36314293) where there is a full description of the participant numbers and intervention as well as descriptive tables on participant characteristics and results. The comment about discussing lessons learned about the behavior interaction and change is an excellent one. The ACT1ON Study did collect extensive surveys including Ingestive Behavior & Nutrition Knowledge, Diabetes Eating Problem Survey, Dutch Eating Behavior Questionnaire, General Impulsivity Questionnaire, Good Craving Inventory, Weight-Related Eating Questionnaire, Nutrition Knowledge Questionnaire, Diabetes-Related Quality of Life, Fear of Hypoglycemia and Acceptability of Experimental Diets. This survey data is very robust and is still being analyzed and will be included in future manuscripts. Lastly, more details were added regarding the in-person and Zoom registered dietitian (RD) education sessions based on interviewing (MI) and problem-solving skills training (PSST) to teach practical problem-solving tailored to the patient by way of a multi-faceted intervention, supplemented by a flexible array of tools (see Part 1.4, page 5). To avoid replicating the pilot findings from the main ACT1ON Study manuscript, please refer to Igudesman et al. Weight management in young adults with type 1 diabetes: The advancing care for type 1 diabetes and obesity network sequential multiple assignment randomized trial pilot results. Diabetes Obes Metab, 2023 (PMID: 36314293) for the full study summary.

Added text: ACT1ON participants met with Registered Dietitians (RD) at the intervention introduction in-person or on Zoom monthly as well as phone check-ins utilizing the above described theories and approaches. Additional sessions with the RD also occurred during diet re-randomization at 3 and 6 months33 (Figure 3).

And, “These findings confirm what is well-recognized clinically, which is that “one size does not fit all” and the heterogeneity of weight loss responses to diet for people with T1D are complex and likely influenced by genetic, metabolic, behavioral, and environmental factors33.  However, this trial did demonstrate that short-term weight loss is achievable for those with T1D while sustaining or improving HbA1c without increasing hypoglycemia33.”

Added text: As such, research studies aimed at assessing the effects of diet alone, PA alone, and the combination of diet plus PA on weight management in T1D are warranted.”

  1. The conclusion in the text should be improved, since it only summarized the role of PA, then what is the conclusion for diet in this article?

Thank you for this feedback. We have expanded the conclusion section and added more details regarding diet in Part 4, page 8.

Added text: “There is no “one size fits all diet” or specific eating pattern for people with T1D and diet plans should be individualized to the person with TID11,12. ACTION did demonstrate that short-term weight loss is achievable for those with T1D while sustaining or improving HbA1c without increasing hypoglycemia33.”

Reviewer 2 Report

This manuscript review dietary strategies, the role of combined PA and diet for weight management, current resources for PA and glucose management, barriers to PA adherence in adults with T1D, as well as findings and lessons learned from the ACT1ON. It is generally well-written, but there are some comments that the authors need to address.

Major and minor comments:

1. The background is insufficient, the role of diet or combined PA and diet for weight management should also be introduced.

2. Page2, Part1.2, I agree that studies investigating the combined effects of PA and diet in T1D are still limited, but the combined effects of PA and diet on weight management in other subjects has been studied sufficiently, such as adolescents with obesity. While, to my opinion, the comparison of the effects of diet alone, PA alone, and diet plus PA on human weight management are still limited. So, it maybe necessary to address this point and worthy to be investigated in your following research.

3. The conclusion in the text should be improved, since it only summarized the role of PA, then what is the conclusion for diet in this article?

Author Response

May 17, 2023

Manuscript ID: nutrients-2382218
Type of manuscript: Article
Title: An overview of diet and physical activity for healthy weight in 
adolescents and young adults with type 1 diabetes: Lessons learned from the 
ACTION consortium
Authors: Franziska K Bishop *, Ananta Addala, Karen D. Corbin, Franklin R. 
Muntis, Richard E. Pratley, Michael C. Riddell, Elizabeth J Mayer-Davis, 
David M Maahs, Dessi P. Zaharieva
Received: 19 April 2023
E-mails: fbishop@stanford.edu, aaddala@stanford.edu, 
karen.corbin@adventhealth.com, frmuntis@email.unc.edu, 
richard.pratley.md@adventhealth.com, mriddell@yorku.ca, 
mayerdav@email.unc.edu, dmaahs@stanford.edu, dessi@stanford.edu
Submitted to section: Pediatric Nutrition,
https://www.mdpi.com/journal/nutrients/sections/Pediatric_Nutrition
Nutrition Managing in Pediatric Diabetes: Aspects and Challenges
https://www.mdpi.com/journal/nutrients/special_issues/pediatricdiabetes_nutrition

Responses to Reviewers

Reviewer 2:

This manuscript review dietary strategies, the role of combined PA and diet for weight management, current resources for PA and glucose management, barriers to PA adherence in adults with T1D, as well as findings and lessons learned from the ACT1ON. It is generally well-written, but there are some comments that the authors need to address.

Major and minor comments:

Responses in bold. 

  1. The background is insufficient, the role of diet or combined PA and diet for weight management should also be introduced.

Thank you for this feedback. We have included the role of diet for weight management in the revised manuscript and the role of diet and PA weight management in people with T1D has also been introduced in the introduction in Part 1, page 2.

Added text: “While there is no single diet or specific eating pattern for people with T1D, metabolic status, clinical goals, and personal preferences, and sociocultural factors should always be considered11. In addition, existing evidence shows no ideal macronutrient distribution for weight loss12 and diet plans should be individualized for all youth with diabetes. Carbohydrate/energy intake recommendations in older, overweight, or obese adolescents may be lower (~40% energy) with higher protein intake (~25% energy) according to clinical consensus12.

And, “Barriers to PA and diet in those with TID include less metabolic flexibility during exercise24–28, fear of exercise-associated hypoglycemia29, and lack of specific recommendations for weight management for people with T1D30. However, with a lack of evidence and research on obesity treatment in people with T1D, there is currently limited evidence to base recommendations on30.”

  1. Page2, Part1.2, I agree that ‘studies investigating the combined effects of PA and diet in T1D are still limited’,but the combined effects of PA and diet on weight management in other subjects has been studied sufficiently, such as adolescents with obesity. While, to my opinion, the comparison of the effects of diet alone, PA alone, and diet plus PA on human weight management are still limited. So, it maybe necessary to address this point and worthy to be investigated in your following research.

Thank you for this point. We agree that there is more published literature on the combined effects of PA and diet on weight management and that this area is still very much understudied in the T1D population. As such, we have added more context and highlighted this point in Part 1.2, pages 2-3.

Added text: As such, research studies aimed at assessing the effects of diet alone, PA alone, and the combination of diet plus PA on weight management in T1D are warranted.”

  1. The conclusion in the text should be improved, since it only summarized the role of PA, then what is the conclusion for diet in this article?

Thank you for this feedback. We have expanded the conclusion section and added more details regarding diet in Part 4, page 8.

Added text: “There is no “one size fits all diet” or specific eating pattern for people with T1D and diet plans should be individualized to the person with TID11,12. ACTION did demonstrate that short-term weight loss is achievable for those with T1D while sustaining or improving HbA1c without increasing hypoglycemia33.”

Round 2

Reviewer 2 Report

None